# Zero-Shot Video Question Answering via Frozen Bidirectional Language Models

**Antoine Yang**[1,2], **Antoine Miech**[3], **Josef Sivic**[4], **Ivan Laptev**[1,2], **Cordelia Schmid**[1,2]
[1]Inria Paris    [2]Département d'informatique de l'ENS, CNRS, PSL Research University
[3]DeepMind    [4]CIIRC CTU Prague
https://antoyang.github.io/frozenbilm.html

## Abstract

Video question answering (VideoQA) is a complex task that requires diverse multi-modal data for training. Manual annotation of questions and answers for videos, however, is tedious and prohibits scalability. To tackle this problem, recent methods consider zero-shot settings with no manual annotation of visual question-answer. In particular, a promising approach adapts *frozen autoregressive* language models pretrained on Web-scale text-only data to multi-modal inputs. In contrast, we here build on *frozen bidirectional* language models (BiLM) and show that such an approach provides a stronger and cheaper alternative for zero-shot VideoQA. In particular, (i) we combine visual inputs with the frozen BiLM using light trainable modules, (ii) we train such modules using Web-scraped multi-modal data, and finally (iii) we perform zero-shot VideoQA inference through masked language modeling, where the masked text is the answer to a given question. Our proposed approach, *FrozenBiLM*, outperforms the state of the art in zero-shot VideoQA by a significant margin on a variety of datasets, including LSMDC-FiB, iVQA, MSRVTT-QA, MSVD-QA, ActivityNet-QA, TGIF-FrameQA, How2QA and TVQA. It also demonstrates competitive performance in the few-shot and fully-supervised setting. Our code and models are publicly available at [1].

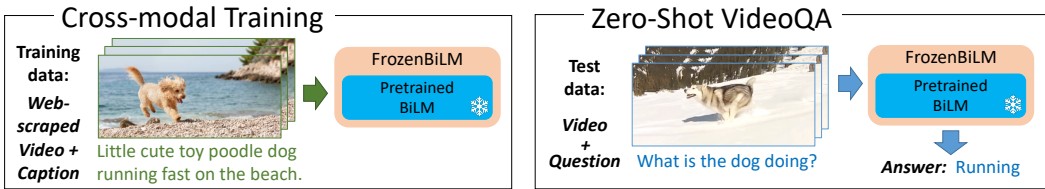

Figure 1: Our model *FrozenBiLM* builds on a pretrained and *frozen* bidirectional language model (BiLM), and is trained from Web-scraped video-caption pairs. *FrozenBiLM* excels in the zero-shot video question answering task without using any explicit visual question-answer supervision.

## 1 Introduction

Video question answering (VideoQA) is a challenging task that requires fine-grained multi-modal understanding. State-of-the-art approaches to VideoQA [43, 107, 109] rely on large video datasets manually annotated with question-answer pairs. Yet, collecting such annotations is time consuming, expensive and therefore not scalable. This has motivated the development of *zero-shot* VideoQA approaches [101, 102, 110], that use no visual question-answer annotation for training, see Figure 1.

---

[4]Czech Institute of Informatics, Robotics and Cybernetics at the Czech Technical University in Prague.

36th Conference on Neural Information Processing Systems (NeurIPS 2022).

Recently, a promising line of work builds on *frozen* large autoregressive language models [19, 68, 91, 96, 104, 111] for zero-shot visual question answering. This has been motivated by the findings from GPT-3 [8] which exhibits strong zero-shot text-only question answering abilities from large autoregressive language models. Such models [8, 72, 82, 92] can predict an arbitrarily long sequence of text, one token at each step from left to right. However, they usually require billion parameters to work well, making them computationally expensive to train, and challenging to deploy in practice.

In contrast, recent work in natural language [65, 76, 77, 87] demonstrates strong zero-shot performance for lighter bidirectional language models (BiLM). Such models [17, 25, 35, 42, 61, 75] can predict a few masked tokens in an input sequence given left and right context in a single forward pass. These works cast downstream tasks in *cloze* form[1] [90], similar to the masked language modeling task (MLM) [17] solved by these models at pretraining. This motivates us to tackle diverse zero-shot multi-modal tasks (open-ended VideoQA [98], multiple-choice VideoQA [46] and fill-in-the-blank [66]) by formulating them in *cloze* form and leveraging the text-only knowledge of pretrained BiLM.

To adapt a pretrained BiLM to multi-modal inputs, we combine it with a frozen pretrained visual backbone and a set of lightweight additional modules including adapters [28]. We train these modules on Web-scraped video-text data using a simple visually-conditioned MLM loss. We preserve the uni-modal knowledge of a BiLM by *freezing* its weights. To our knowledge, our approach is the first to explore the zero-shot visual-linguistic capabilities of *frozen non-autoregressive* language models.

We show that our approach largely improves the state of the art on various zero-shot VideoQA benchmarks. Furthermore, we demonstrate that *frozen bidirectional* language models perform better while being cheaper to train than *frozen autoregressive* language models [91]. Moreover, our ablation studies show (i) the ability of our model to effectively perform zero-shot multi-modal reasoning using both visual cues and speech transcripts, (ii) the importance of adapters combined with *frozen* pretrained language models, (iii) the impact of multi-modal data scale, (iv) the impact of the language model size and of bidirectional modeling. Our approach also performs competitively in the fully-supervised setting. Indeed, we show the benefits of *freezing* the weights of a BiLM when using VideoQA training data, while updating considerably less parameters compared to alternative methods. Finally, we introduce a new few-shot VideoQA task in which we finetune our pretrained model on a small fraction of the downstream training dataset, and show promising results in this setting.

In summary, our contributions are three-fold:
- *(i)* We present *FrozenBiLM*, a framework that handles multi-modal inputs using *frozen* bidirectional language models and enables zero-shot VideoQA through masked language modeling.
- *(ii)* We provide an extensive ablation study and demonstrate the superior performance of our framework in the zero-shot setting when compared to previous autoregressive models.
- *(iii)* Our approach improves the state of the art in zero-shot VideoQA by a significant margin. *FrozenBiLM* also demonstrates competitive performance in the fully-supervised setting and shows strong results in the few-shot VideoQA setting which we introduce.

Our code and trained models are publicly available at [1].

## 2   Related Work

**Zero-shot VideoQA.** A vast majority of VideoQA approaches rely on relatively small, manually annotated VideoQA datasets [3, 9, 10, 13–15, 20, 23, 24, 30, 33, 34, 36–39, 43, 44, 47, 58, 60, 69, 70, 74, 78, 79, 83, 89, 97, 100, 103, 105, 112, 116]. Recently, a few work [101, 110] have explored zero-shot approaches for VideoQA, where models are *only* trained on automatically mined video clips with short text descriptions. In contrast to VideoQA annotations, such video-text pairs are readily-available at scale on the Web [6, 67, 109]. In particular, Yang et al. [101] automatically generate VideoQA training data using language models [72] pretrained on a manually annotated text-only question-answer corpus [73]. Reserve [110] uses GPT-3 [8] to rephrase questions into sentences completed by a multi-modal model. In contrast to these prior works [101, 110], our method does not require any kind of explicitly annotated language dataset or the use of data generation pipelines for zero-shot VideoQA. Note that BLIP [53] studies a related setting where a model trained on manually annotated image-question-answer triplets is transferred to VideoQA, which is a less challenging task. Also note that VideoCLIP [99] considers a related zero-shot multiple-choice video-to-text retrieval task as VideoQA, but in this setting the model is not provided with natural language questions.

---

[1]"Cloze test" is an exercise test where certain portions of text are occluded or masked and need to be filled-in.

**Visual language models.** As language models require large amounts of training data to perform well [27], recent works have studied transferring pretrained language models [8, 94] to image-text tasks. VisualGPT [11] and VC-GPT [64] showed the benefit of initializing the weights of an image captioning model with a pretrained autoregressive language-only model. Recent work pushed this idea further by *freezing* the weights of a pretrained autoregressive language model for tackling vision and language tasks [2, 19, 68, 91, 96, 104, 111]. Our approach also leverages a *frozen* pretrained language model. Similar to MAGMA [19], we also use adapter layers [28, 29]. However, we differ from these approaches as we propose to instead use lighter *bidirectional masked language models*, instead of autoregressive ones, and rely on a masked language modeling objective (MLM) instead of an autoregressive one. Moreover, our model is specifically designed for videos, for which high-quality visual question answering annotation is even more scarce compared to still images [19, 68, 91, 104]. We also explore the use of the speech modality, and tackle tasks which are challenging for autoregressive language models such as video-conditioned fill-in-the-blank [66]. Finally we show in Section 4.3 the superior performance of frozen bidirectional language models in comparison with autoregressive ones [91].

**Masked Language Modeling in vision and language.** The MLM objective was initially introduced in natural language [17, 42, 61] to pretrain bidirectional transformers and learn generic representations. This approach achieved state-of-the-art results in many language tasks after finetuning on downstream datasets. Its success inspired numerous works to adapt it to train multi-modal transformer models on paired visual-linguistic data [12, 21, 22, 26, 31, 40, 48, 51, 56, 54, 59, 52, 50, 62, 63, 80, 81, 85, 86, 88, 93, 95, 106, 109, 114, 115]. However, these works typically use it to learn generic visual-linguistic representations by updating the transformer weights, and then use expensive manual supervision to train randomly initialized task-specific answer classifiers for VQA [12, 22, 51, 52, 59, 62, 80, 81, 85, 88, 95, 106] or VideoQA [21, 48, 50, 93, 109]. In contrast, we tackle *zero-shot* VideoQA, *i.e.* without using *any* manual annotation. Moreover, we do not update the transformer weights during cross-modal training, but instead exhibit the benefits of *freezing* these weights after text-only pretraining, for both zero-shot and fully-supervised VideoQA (see Sections 4.2 and 4.5).

# 3 Method

This section presents our approach to tackle *zero-shot* video question answering. Here, zero-shot means that we do not use *any* visual question answering annotation and only rely on scalable data from the Web. Our approach starts with two strong pretrained components: (i) a text-only bidirectional masked language model (BiLM) pretrained on data from the Internet, which has the capability of zero-shot question answering but is not capable of visual reasoning, and (ii) a vision encoder pretrained to map images to text descriptions, but which does not have the ability to perform visual question answering. We aim at connecting these two components while keeping the language component *frozen* to avoid catastrophic forgetting [16], where the large language model would specialize to a new task while forgetting its initial capabilities. The end-goal is to design a unified model having the best of both worlds: visual understanding capabilities of a powerful visual encoder and question answering capabilities of a powerful language model. This requires several technical innovations, which are described in the rest of this section. First, we explain in Section 3.1 how we augment a *frozen* pretrained bidirectional masked language model with new layers to enable joint video and language reasoning, see Figure 2. Second, we present in Section 3.2 how we train these layers on video-text data scraped from the Web [6]. Finally, we describe in Section 3.3 how we enable zero-shot predictions for several video-language downstream tasks, including open-ended VideoQA, by casting them in a *cloze* form, similar to the masked language modeling task solved during training.

## 3.1 Architecture

The proposed architecture, illustrated in Figure 2, brings together a powerful *frozen* pretrained bidirectional language model with a strong visual encoder. The difficulty lies in enabling multi-modal reasoning while keeping the large language model *frozen*. To address this challenge, we unify these two models via a visual-to-text projection module together with small adapter modules inserted within the frozen language model. Next, we describe in more detail the three main components of the architecture: (i) the *frozen* pretrained bidirectional language model, (ii) the pretrained video encoder and (iii) the lightweight modules that seamlessly connect the two components.

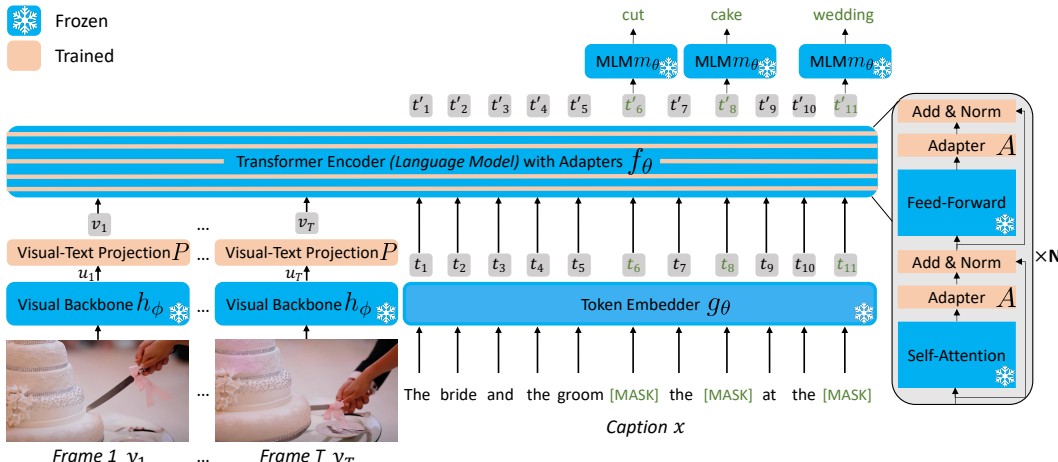

Figure 2: **Our training architecture** consists of a large *frozen* bidirectional language model (BiLM) and a *frozen* pretrained visual encoder (in blue), complemented with additional lightweight trainable modules (in orange): (1) a visual-to-text projection module $P$ (on the left), which maps the *frozen* visual features to the joint visual-text embedding space and (2) a set of small adapter modules $A$ (on the right) in between the *frozen* transformer blocks. The pretrained normalization layers in the BiLM (on the right) are also finetuned.

***Frozen* Bidirectional Masked Language Model.** Our method starts from a pretrained bidirectional language model based on a Transformer encoder [92]. The input text is decomposed into a sequence of tokens $x = \{x_i\}_1^L \in [1, V]^L$ by a tokenizer of a vocabulary size $V$. The language model, parameterized by $\theta$, makes use of an embedding function $g_\theta$ which independently transforms each token into a $D$-dimensional continuous embedding $t = \{t_i\}_1^L := \{g_\theta(x_i)\}_1^L \in \mathbb{R}^{L \times D}$, a Transformer encoder $f_\theta$ which computes interactions between all input tokens and outputs contextualized representations $t' = \{t'_i\}_1^L$, and a masked language modeling (MLM) classifier head $m_\theta$ which independently maps the $D$-dimensional continuous embedding for each token $t'_i$ to a vector of logits parameterizing a categorical distribution over the vocabulary $V$. This distribution is referred to by $\log p_\theta(x) := \{m_\theta(t'_i)\}_1^L \in \mathbb{R}^{L \times V}$. We assume that the language model is pretrained, *i.e.* $\theta$ has been optimised with a standard MLM objective [17] on a large dataset of text from the Web. We show in Section 4.2 that this text-only pretraining has a crucial importance for zero-shot VideoQA.

**Pretrained Video Encoder.** The video is represented by a sequence of frames $y = \{y_i\}_1^T$. Each frame is forwarded separately through a visual backbone $h_\phi$, which outputs one feature vector per frame $u = \{u_i\}_1^T := \{h_\phi(y_i)\}_1^T \in \mathbb{R}^{T \times D_u}$. In detail, the visual backbone is CLIP ViT-L/14 [18, 71] at resolution $224 \times 224$ pixels, pretrained to map images to text descriptions with a contrastive loss on 400M Web-scraped image-text pairs. The backbone is kept frozen throughout our experiments. Note that a CLIP-baseline for zero-shot VideoQA results in poor performance, see Section 4.4.

**Connecting the Frozen Language and Frozen Vision components.** The video features are incorporated into the language model as a prompt [49, 57, 113] $v$ of length $T$ (Figure 2, left). This prompt is obtained by linearly mapping the visual features $u$ to the text token embedding space via a visual-to-text projection $P \in \mathbb{R}^{D_u \times D}$, *i.e.* $v = \{v_i\}_1^T := \{P(u_i)\}_1^T$. The prompt is then concatenated with the text embeddings before being forwarded to the transformer encoder that models joint visual-linguistic interactions. We show in Section 4.2 that incorporating the input video considerably improves zero-shot VideoQA results. In addition, to learn powerful multi-modal interactions while keeping the transformer encoder weights *frozen*, we equip the transformer encoder with lightweight adapter modules $A$ [28] (Figure 2, right). We use an adapter which transforms the hidden state $z$ with a multi-layer perceptron transformation and a residual connection, *i.e.* $A(z) = z + W^{up}\psi(W^{down}z)$ with $W^{down} \in \mathbb{R}^{D \times D_h}$, $W^{up} \in \mathbb{R}^{D_h \times D}$, $D$ the hidden dimension of the transformer, $D_h$ the bottleneck dimension, and $\psi$ a ReLU activation function. $D_h$ is typically set to be smaller than $D$ such that the adapters are lightweight. In detail, we add an adapter module before the layer normalization, after each self-attention layer and each feed-forward layer of the transformer encoder.

## 3.2 Cross-modal training

We wish to train the newly added modules introduced in the previous section (shown in orange in Figure 2) for the VideoQA task. This is hard because we assume that no explicit manual annotation for the VideoQA task is available, such annotations being expensive and therefore hard to obtain at scale. Instead we train our architecture using *only* readily-available video-caption pairs scraped from the Web. Such data is easy to obtain [6, 67, 109], ensuring the scalability of our approach.

During training, we keep the weights of the pretrained BiLM and pretrained visual backbone *frozen* as previously explained. We train from scratch the parameters of (i) the visual-to-text projection module $P$ and (ii) the adapter modules $A$. We show in Section 4.2 the importance of *freezing* the BiLM weights combined with training the adapter modules. Note that all normalization layers [5] of the pretrained BiLM are also updated to adjust to the new distribution of the training data. We denote all the trainable parameters of our model by the subscript $\mu$. In practice, they sum up to about 5% of the BiLM parameters, hence the training of our model is computationally efficient.

We use a visually-conditioned masked language modeling objective (MLM), in which some text tokens $\{x_m\}$ are randomly masked and the model has to predict these tokens based on the surrounding text tokens and the video input. Formally, we minimize the following loss:

$$\mathcal{L}_\mu(x, y) = -\frac{1}{M} \sum_m \log p_\mu(\tilde{x}, y)_m^{x_m},\tag{1}$$

where $\tilde{x}$ is the corrupted text sequence, $y$ is the sequence of video frames, $p_\mu(\tilde{x}, y)_m^{x_m}$ is the probability for the (masked) $m$-th token in $\tilde{x}$ to be $x_m$, and $M$ is the number of masks in the sequence $\tilde{x}$. In detail, we follow [17] and corrupt 15% of text tokens, replacing them 80% of the time with a mask token, 10% of the time with the same token and 10% of the time with a randomly sampled token.

## 3.3 Adapting to downstream tasks

After training, our model is able to fill gaps in the input text given an input video together with left and right textual context as part of the input text. We wish to apply our model *out-of-the-box* to predict an answer given a question about a video. The video can optionally come with textual subtitles obtained using automatic speech recognition. To avoid using manual supervision, we formulate the downstream tasks in *cloze* form [76, 90], *i.e.* such that the model only has to fill-in a mask token in the input prompt similarly to the MLM objective optimized during training. The adaptation to the downstream tasks brings several challenges, as described next. First, we describe how we formulate the input text prompts for several downstream tasks. Then, we explain how we map the mask token from the input text prompt to an answer via a *frozen* answer embedding module. Finally, we present how we finetune our architecture in a supervised setting.

**Input prompt engineering.** We describe how we design the input text prompts for several downstream video-language tasks. Each downstream task is formulated as a masked language modeling problem. This allows us to apply *FrozenBiLM* out-of-the-box. A [CLS] token and a [SEP] token are respectively inserted at the start and the end of each sequence following [17].

*Open-ended VideoQA.* Given a question and a video, the task is to find the correct answer in a large vocabulary $\mathcal{A}$ of about 1K answers. Answers are concise, *i.e.* the great majority of answers consist of one word [32, 98, 101, 108]. We design the following prompt:
``[CLS] Question: <Question>? Answer: [MASK]. Subtitles: <Subtitles> [SEP]''

*Multiple-choice VideoQA.* Given a question and a video, the task is to find the correct answer in a small number of candidates $C$, typically up to 5 choices [46, 54]. We set the vocabulary to $\mathcal{A} = [\text{Yes}, \text{No}]$ and compute a confidence score for each candidate by using the following prompt:
``[CLS] Question: <Question>? Is it '<Answer Candidate>'? [MASK]. Subtitles: <Subtitles> [SEP]''
We choose the best option by selecting the candidate with the highest *Yes* logit value.

*Video-conditioned fill-in-the-blank task.* Given a video and a sentence with a blank space, the task is to fill in the blank with the correct word from a vocabulary $\mathcal{A}$ of about 1K answers. We replace the blank in the sentence with a mask token, and design the following prompt:
``[CLS] <Sentence with a [MASK] token>. Subtitles: <Subtitles> [SEP]''

Note that all prompts are prepended with the video prompt (see Section 3.1) before being forwarded to the transformer encoder.

**Answer embedding module.** For each downstream task, we wish to map the mask token in the input text prompt to an actual answer prediction in the set of possible answers $\mathcal{A}$, as described above. For this we use the *frozen* MLM classifier head $m_\theta$. However, $m_\theta \in \mathbb{R}^{V \times D}$ covers $V$ different tokens where $V >> N$ and $N \approx 1,000$ is the size of $\mathcal{A}$. Therefore, we introduce a task-specific answer classification head $l$ which linearly maps a contextualized mask representation $t'_i$ to a vector of logits parameterizing a categorical distribution over the vocabulary $\mathcal{A}$, *i.e.* $l \in \mathbb{R}^{N \times D}$. We set the weights of this answer module $l$ with the corresponding weights of the pretrained MLM classifier $m_\theta$ for one-token answers. In the case of multi-token answers, we average the weights of their different tokens. We, hence, enable zero-shot inference at test time. We also discuss other alternative strategies to handle multi-token answers in the Supplementary Material.

**Fully-supervised training.** To evaluate our approach on fully-supervised benchmarks, we also explore finetuning of our model on datasets that provide manual annotations for the target task. To this end, we train the same parameters as explained in Section 3.2, while keeping the transformer weights and the answer embedding module *frozen*. For open-ended VideoQA and video-conditioned fill-in-the-blank, we use a cross-entropy loss on the task-specific vocabulary $\mathcal{A}$. For multiple-choice VideoQA, we use a binary cross-entropy loss applied to each answer candidate. We show in Section 4.5 the benefit of *freezing* the language model weights during fully-supervised training.

## 4 Experiments

This section demonstrates the benefits of our *FrozenBiLM* framework and compares our method to the state of the art. We first outline our experimental setup in Section 4.1. We then present ablation studies in Section 4.2. Next we compare our bidirectional framework to its autoregressive variant in Section 4.3. The comparison to the state of the art in zero-shot VideoQA and qualitative results are presented in Section 4.4. Finally, we finetune our model on the VideoQA task in Section 4.5, where we show few-shot and fully-supervised results.

### 4.1 Experimental setup

**Frozen bidirectional language model.** We use a tokenizer based on SentencePiece [41] with $V = 128,000$, and a bidirectional language model with 900M parameters, DeBERTa-V2-XLarge [25], trained with the MLM objective on a corpus of 160G text data. We also show how our approach generalizes to other MLM-pretrained bidirectional language models such as BERT [17] in Section 4.2.

**Datasets.** For training we use the publicly available **WebVid10M** dataset [6], which consists of 10 million of video-text pairs scraped from the Shutterstock website where video captions are obtained from readily-available alt-text descriptions. We evaluate results on eight downstream datasets covering a wide range of textual and video domains (*e.g.* GIFs, YouTube videos, TV shows, movies), and multiple VideoQA paradigms: open-ended VideoQA (**iVQA** [101], **MSRVTT-QA** [98], **MSVD-QA** [98], **ActivityNet-QA** [108] and **TGIF-QA** FrameQA [32]), multiple-choice VideoQA (**How2QA** [54] and **TVQA** [46]) and video-conditioned fill-in-the-blank (**LSMDC**-Fill-in-the-blank [66]). Unless stated otherwise, we report top-1 test accuracy using the original splits for training, validation and test. For How2QA, we report results on the public validation set for comparison with prior work [78, 101, 107]. For TVQA, we report results on the validation set for the ablation studies and on the hidden test set for the comparison to the state of the art. More details are included in the Supplementary Material.

**Implementation Details.** The training for 2 epochs on WebVid10M lasts 20 hours on 8 Tesla V100 GPUs. We give further details in the Supplementary Material.

### 4.2 Ablation studies

In this section, we evaluate the zero-shot performance of different variants of our method. By default, we use the *frozen* pretrained DeBERTa-V2-XLarge language model and train the visual-to-text-

| | LM Pretraining | Frozen LM | Adapters | Fill-in-the-blank LSMDC | Open-ended | | | | | Multiple-choice | |
|---|---|---|---|---|---|---|---|---|---|---|---|
| | | | | | iVQA | MSRVTT-QA | MSVD-QA | ActivityNet-QA | TGIF-QA | How2QA | TVQA |
| 1. | ✗ | ✗ | ✗ | 0.5 | 0.3 | 0.1 | 0.0 | 0.5 | 0.0 | 32.4 | 20.7 |
| 2. | ✓ | ✗ | ✗ | 37.1 | 21.0 | **17.6** | 31.9 | 20.7 | 30.7 | 45.7 | 45.6 |
| 3. | ✓ | ✓ | ✗ | 50.7 | **27.3** | 16.8 | 32.2 | 24.7 | 41.0 | 53.5 | 53.4 |
| 4. | ✓ | ✓ | ✓ | **51.5** | 26.8 | 16.7 | **33.8** | **25.9** | **41.9** | **58.4** | **59.2** |

Table 1: The effect of initializing and training various parts of our model evaluated on zero-shot VideoQA. All models are trained on WebVid10M and use multi-modal inputs (video, speech and question) at inference.

| | Visual | Speech | Fill-in-the-blank LSMDC | Open-ended | | | | | Multiple-choice | |
|---|---|---|---|---|---|---|---|---|---|---|---|
| | | | | iVQA | MSRVTT-QA | MSVD-QA | ActivityNet-QA | TGIF-QA | How2QA | TVQA |
| 1. | ✗ | ✗ | 47.9 | 11.0 | 6.4 | 11.3 | 22.6 | 32.3 | 29.6 | 23.2 |
| 2. | ✗ | ✓ | 49.8 | 13.2 | 6.5 | 11.7 | 23.1 | 32.3 | 45.9 | 44.1 |
| 3. | ✓ | ✗ | 50.9 | 26.2 | **16.9** | 33.7 | **25.9** | **41.9** | 41.9 | 29.7 |
| 4. | ✓ | ✓ | **51.5** | **26.8** | 16.7 | **33.8** | **25.9** | **41.9** | **58.4** | **59.2** |

Table 2: Impact of the visual and speech modalities on zero-shot VideoQA. Rows 1 and 2 report results for a pretrained language model without any visual input. Rows 3 and 4 give results for a *FrozenBiLM* model pretrained on WebVid10M.

projection layer together with adapters for 2 epochs on WebVid10M. We refer to this default model as *FrozenBiLM*. This model uses three input modalities in terms of video, question, and speech.

**Ablation of the model training.** We ablate the effect of initializing parameters of the language model, freezing its weights and training adapters in Table 1. We observe that the language model pretraining is crucial. Indeed, a model with randomly initialized language weights (row 1) performs poorly compared to models initialized with language pretrained weights (rows 2 to 4). Moreover, the model which updates the language model weights (row 2) during cross-modal training performs considerably worse compared to variants that *freeze* them (rows 3 and 4). This shows the benefit of *freezing* the language model for zero-shot VideoQA. We also notice the benefit of the adapter layers by comparing rows 3 and 4, especially for multiple-choice datasets. Finally, we note that training variants with the *frozen* language model is twice faster compared to updating all parameters, as there is a significantly lower number of parameters to be trained.

**Impact of modalities.** Table 2 shows the impact of the visual and speech modalities on the zero-shot performance of our model. First, we evaluate the text-only performance of our model using neither visual input nor speech input in row 1. We can observe that adding speech (row 2) marginally improves the results and that the importance of speech highly depends on the dataset. When adding vision (rows 3 and 4), the performance increases significantly, *e.g.* +13.6% accuracy on iVQA and +22.1% on MSVD-QA between rows 4 and 2. Finally, the model with vision also benefits from the speech, *e.g.* +16.5% accuracy on How2QA and +29.5% accuracy on TVQA (compare rows 3 and 4).

Note that in practice, speech is missing for many videos, as we obtain the speech directly from the YouTube API and many videos are no longer available. Exceptions are How2QA and TVQA for which the authors [46, 55] provide speech for all videos. Consequently, we have speech data for only 44.3%, 14.2%, 8.2%, 7.1% and 25.3% of test samples in LSMDC-FiB, iVQA, MSRVTT-QA, MSVD-QA and ActivityNet-QA respectively. GIFs in TGIF-QA do not contain speech.

**Size of the cross-modal training dataset.** Zero-shot results of *FrozenBiLM* after training for a fixed number of iterations on different fractions of WebVid10M are shown in Table 3. We construct these subsets such that larger subsets include the smaller ones. We find that performance increases monotonically with more multi-modal training data.

| | Training Data | MSVD-QA | How2QA |
|---|---|---|---|
| 1. | WebVid1K | 13.6 | 53.0 |
| 2. | WebVid10K | 22.7 | 54.9 |
| 3. | WebVid200K | 27.8 | 56.0 |
| 4. | WebVid2M | 30.1 | 57.4 |
| 5. | WebVid10M | **33.8** | **58.4** |

Table 3: Dependency on the size of the training set. Zero-shot results are presented for different fractions of the WebVid10M dataset used for training.

**Size of the language model.** In Table 4, we ablate the importance of the language model size for the zero-shot performance. Note that when comparing different language models, we use no adapters to avoid biases related to the choice of the bottleneck dimension hyperparameter [28]. We find that using the 900M-parameter DeBERTA-V2-XLarge (row 6) outperforms the 300M-parameter BERT-Large (row 5) which also improves over the 100M-parameter BERT-Base (row 4).

| Method | Language Model | # LM params | Train time (GPUH) | iVQA | MSRVTT-QA | MSVD-QA | ActivityNet-QA | TGIF-QA |
|---|---|---|---|---|---|---|---|---|
| Autoregressive | 1. GPT-Neo-1.3B | 1.3B | 200 | 6.6 | 4.2 | 10.1 | 17.8 | 14.4 |
| | 2. GPT-Neo-2.7B | 2.7B | 360 | 9.1 | 7.7 | 17.8 | 17.4 | 20.1 |
| | 3. GPT-J-6B | 6B | 820 | 21.4 | 9.6 | 26.7 | 24.5 | 37.3 |
| Bidirectional | 4. BERT-Base | **110M** | **24** | 12.4 | 6.4 | 11.7 | 16.7 | 23.1 |
| | 5. BERT-Large | 340M | 60 | 12.9 | 7.1 | 13.0 | 19.0 | 21.5 |
| | 6. DeBERTa-V2-XLarge | 890M | 160 | **27.3** | **16.8** | **32.2** | **24.7** | **41.0** |

Table 4: Comparison of autoregressive language models (top) and bidirectional language models (bottom) for zero-shot VideoQA. All variants are trained on WebVid10M for the same number of epochs.

| Method | Training Data | Speech | Fill-in-the-blank LSMDC | Open-ended | | | | | Multiple-choice | |
|---|---|---|---|---|---|---|---|---|---|---|
| | | | | iVQA | MSRVTT-QA | MSVD-QA | ActivityNet-QA | TGIF-QA | How2QA | TVQA |
| Random | — | — | 0.1 | 0.1 | 0.1 | 0.1 | 0.1 | 0.1 | 25 | 20 |
| CLIP ViT-L/14 [71] | 400M image-texts | ✗ | 1.2 | 9.2 | 2.1 | 7.2 | 1.2 | 3.6 | 47.7 | 26.1 |
| Just Ask [102] | HowToVQA69M + WebVidVQA3M | ✗ | — | 13.3 | 5.6 | 13.5 | 12.3 | — | 53.1 | — |
| Reserve [110] | YT-Temporal-1B | ✗ | 31.0 | — | 5.8 | — | — | — | — | — |
| *FrozenBiLM* (Ours) | WebVid10M | ✗ | 50.9 | 26.2 | 16.9 | 33.7 | 25.9 | 41.9 | 41.9 | 29.7 |
| *FrozenBiLM* (Ours) | WebVid10M | ✓ | **51.5** | **26.8** | 16.7 | **33.8** | 25.9 | 41.9 | **58.4** | **59.7** |

Table 5: Comparison with the state of the art for zero-shot VideoQA.

**Importance of the suffix.** Our text input prompts include a suffix just to the right of the mask token which consists in a point and an end-of-sentence token for the variant without speech (or a point followed by the speech subtitles for the variant with speech). We found that removing this suffix leads to a considerable drop of performance (*e.g.* the test accuracy on MSVD-QA in the row 3 of Table 2 drops from 33.7% to 2.8%). Note that we do not observe such a large drop in performance when removing the [CLS] token *e.g.* the accuracy on MSVD-QA drops only from 33.8% to 33.2%. This shows that the bidirectional nature of our framework is a key factor for the performance. Intuitively, this suffix forces the model to provide a concise answer. Such a hard constraint cannot be given to unidirectional autoregressive models compared next in Section 4.3. We further ablate the importance of the prompt design in the Supplementary Material.

## 4.3 Comparison with frozen autoregressive models

In this section, we compare our bidirectional framework using language models of various sizes to the larger, autoregressive GPT-based counterparts recently used for zero-shot image question answering [91, 104]. For fair comparison, we adapt autoregressive models to video and language inputs similarly as our bidirectional models. In detail, autoregressive variants train a similar visual-to-text projection by using a left-to-right language modeling loss [91]. All models in our comparison are trained on WebVid10M for the same number of epochs. At inference, autoregressive variants use the same template as [91] to which we prepend speech subtitles, greedily decode sequences as [91], and use the same answer vocabulary as bidirectional models. Autoregressive variants select the top answer that maximizes the log-likelihood when appended to the question prompt. Here also, we use no adapters for all models, such that the architecture of autoregressive models closely follows [91]. This is to avoid biases related to the tuning of the bottleneck reduction hyperparameter in the adapters [28].

We compare autoregressive and bidirectional language models in terms of accuracy and efficiency in Table 4. We observe that our bidirectional framework (rows 4-6) achieves significantly better zero-shot performance-efficiency trade-off compared to its autoregressive counterpart (rows 1-3). For instance, our framework with BERT-Base [17] (row 4) outperforms the autoregressive variant based on GPT-Neo-1.3B [7] (row 1) which uses 12 times more parameters and 8 times more training time. Likewise, our framework with DeBERTa-V2-XLarge [25] (row 6) improves over the autoregressive variant based on GPT-J-6B [94] (row 3) that has 7 times more parameters and requires 5 times more training time, showing the efficiency of our *bidirectional* framework for zero-shot VideoQA.

## 4.4 Comparison to the state of the art for zero-shot VideoQA

**Quantitative comparison.** Table 5 presents results of our method in comparison to the state of the art in *zero-shot* VideoQA settings [101], *i.e.* when using no manually annotated visual data for training. Our approach outperforms previous methods by a significant margin on all 8 datasets. In

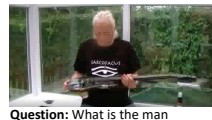 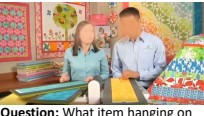 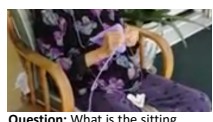 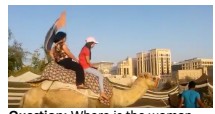 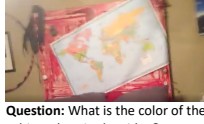

**Question:** What is the man holding at the start of the video?
**GT Answer:** guitar, electric guitar
**Just Ask:** typewriter
**UnFrozenBiLM:** beer
**FrozenBiLM (text-only):** scissors
**FrozenBiLM (ours):** guitar

**Question:** What item hanging on the wall features a tree?
**GT Answer:** quilt
**Just Ask:** christmas sock
**UnFrozenBiLM:** fabric
**FrozenBiLM (text-only):** tree
**FrozenBiLM (ours):** quilt

**Question:** What is the sitting man doing?
**GT Answer:** knit sweater
**Just Ask:** tie cow
**UnFrozenBiLM:** swimming
**FrozenBiLM (text-only):** eating
**FrozenBiLM (ours):** knit sweater

**Question:** Where is the woman sitting on?
**GT Answer:** camel
**Just Ask:** horse yard
**UnFrozenBiLM:** desert
**FrozenBiLM (text-only):** chair
**FrozenBiLM (ours):** camel

**Question:** What is the color of the cabinet door in the video?
**GT Answer:** red
**Just Ask:** dresser
**UnFrozenBiLM:** blue
**FrozenBiLM (text-only):** black
**FrozenBiLM (ours):** red

Figure 3: **Zero-Shot VideoQA.** Qualitative comparison between Just Ask [102] (row 3 in Table 5), our model (row 4 in Table 5), its *unfrozen* variant (row 2 in Table 1) and its text-only variant (row 2 in Table 2). The first two examples are from iVQA [101] and the last three examples are from ActivityNet-QA [108].

| Method | # Trained Params | Fill-in-the-blank LSMDC | Open-ended | | | | | Multiple-choice | |
|---|---|---|---|---|---|---|---|---|---|
| | | | iVQA | MSRVTT-QA | MSVD-QA | ActivityNet-QA | TGIF-QA | How2QA | TVQA |
| HCRN [45] | 44M | — | — | 35.4 | 36.8 | — | 57.9 | — | 71.4* |
| HERO [54] | 119M | — | — | — | — | — | — | 74.1* | 73.6* |
| ClipBERT [48] | 114M | — | — | 37.4 | — | — | 60.3 | — | — |
| Just Ask [102] | 157M | — | 35.4 | 41.8 | 47.5 | 39.0 | — | 85.3 | — |
| SiaSamRea [107] | — | — | — | 41.6 | 45.5 | 39.8 | 60.2 | 84.1 | — |
| MERLOT [109] | 223M | 52.9 | — | 43.1 | — | 41.4 | **69.5** | — | 78.7* |
| Reserve [110] | 644M | — | — | — | — | — | — | — | **86.1*** |
| VIOLET [21] | 198M | 53.7 | — | 43.9 | 47.9 | — | 68.9 | — | — |
| All-in-one [93] | 110M | — | — | 46.8 | 48.3 | — | 66.3 | — | — |
| *UnFrozenBiLM* (Ours) | 890M | 58.9* | 37.7* | 45.0* | 53.9* | **43.2*** | 66.9 | **87.5*** | 79.6* |
| *FrozenBiLM* w/o speech (Ours) | **30M** | 58.6 | **39.7** | **47.0** | 54.4 | 43.2 | 68.6 | 81.5 | 57.5 |
| *FrozenBiLM* (Ours) | **30M** | **63.5*** | 39.6* | **47.0*** | **54.8*** | **43.2*** | 68.6 | 86.7* | 82.0* |

Table 6: Comparison with the state of the art, and the variant *UnFrozenBiLM* which does not freeze the language model weight, on fully-supervised benchmarks. * denotes results obtained with speech input.

| | Supervision | Fill-in-the-blank LSMDC | Open-ended | | | | | Multiple-choice | |
|---|---|---|---|---|---|---|---|---|---|
| | | | iVQA | MSRVTT-QA | MSVD-QA | ActivityNet-QA | TGIF-QA | How2QA | TVQA |
| 1. | 0% (zero-shot) | 51.5 | 26.8 | 16.7 | 33.8 | 25.9 | 41.9 | 58.4 | 59.7 |
| 2. | 1% (few-shot) | 56.9 | 31.1 | 36.0 | 46.5 | 33.2 | 55.1 | 71.7 | 72.5 |
| 3. | 10% (few-shot) | 59.9 | 35.3 | 41.7 | 51.0 | 37.4 | 61.2 | 75.8 | 77.6 |
| 4. | 100% (fully-supervised) | **63.5** | **39.6** | **47.0** | **54.8** | **43.2** | **68.6** | **86.7** | **82.0** |

Table 7: Few-shot results, by finetuning *FrozenBiLM* using a small fraction of the downstream training dataset.

particular, *FrozenBiLM* outperforms Reserve [110], which is trained on one billion YouTube video clips jointly with vision, language and sound, Just Ask [102], which uses large-scale automatically generated VideoQA data, and a CLIP baseline [71] matching the text concatenating question and answer to the middle frame of the video. Note that *FrozenBiLM* performs competitively even when using no speech input. Finally, we note that BLIP [53] has a different definition of *zero-shot* where a network finetuned on the image-VQA dataset [4] is evaluated directly on VideoQA datasets. Our Supplementary Material presents results where we outperform BLIP [53] in their settings and also includes an analysis of results by question type. In summary, our evaluation shows the excellent performance of our model in the challenging zero-shot setup.

**Qualitative results.** Figure 3 illustrates qualitative results of zero-shot VideoQA for our *FrozenBiLM* model and compares them to Just Ask [102], as well as to variants of our approach that do not *freeze* the language model (*UnFrozenBiLM*) and use no visual modality (text-only), as evaluated in Section 4.2. We observe that the *unfrozen* variant can predict answers that lack text-only commonsense reasoning, *e.g.* in the third example, it is unlikely that a sitting man is swimming. The text-only variant does have strong language understanding, but makes visually-unrelated predictions. In contrast, consistently with our quantitative results, our model *FrozenBiLM* is able to correctly answer various questions, showing both a strong textual commonsense reasoning and a complex multi-modal understanding. We show additional qualitative results in the Supplementary Material.

### 4.5 Freezing the BiLM is also beneficial in supervised settings

**Fully-supervised VideoQA.** We next present an evaluation in a supervised setup where we finetune *FrozenBiLM* on a downstream VideoQA task. We emphasize that we also keep our pretrained language model weights *frozen* all throughout finetuning. As shown in Table 6, our approach improves the state

of the art on LSMDC-FiB, iVQA, MSRVTT-QA, MSVD-QA, ActivityNet-QA and How2QA. In particular, *FrozenBiLM* outperforms strong recent baselines such as All-in-one [93] on 2/3 datasets, VIOLET [21] on 3/4 datasets and MERLOT [109] on 4/5 datasets. Our approach has significantly less trainable parameters compared to the state of the art [21, 93, 109] as we *freeze* the weights of the pretrained language model. We ablate this major difference in Table 6, and find that our *FrozenBiLM* with the *frozen* language model performs better and trains twice faster compared to *UnFrozenBiLM* where we update the language model during training. This shows that *freezing* the language model is not only beneficial for zero-shot but also in fully-supervised settings, therefore suggesting that our *FrozenBiLM* framework also provides a parameter-efficient solution for VideoQA training. Finally, we note that *FrozenBiLM* performs competitively even without speech input, although speech helps significantly for the performance on LSMDC, How2QA and TVQA.

**Few-shot VideoQA.** The low number of trainable parameters when training *FrozenBiLM* makes it particularly well-suited in the low data regime. To verify this, we explore a few-shot VideoQA setting where we finetune our pretrained model using varying fractions of VideoQA training data. From Table 7 we observe significant improvements over zero-shot when using only 1% of training data. Finally, we show in Supplementary Material that freezing the BiLM highly benefits the few-shot performance, consistently with the results in the zero-shot and fully-supervised settings.

## 5  Conclusion

We have presented *FrozenBiLM*, a framework that extends *frozen* bidirectional language models to multi-modal inputs by training additional modules on Web-scraped data, and that tackles zero-shot VideoQA through masked language modeling. We have provided extensive ablation studies and shown the efficiency of our framework compared to its autoregressive variant. *FrozenBiLM* improves the state-of-the-art zero-shot VideoQA on various datasets, performs competitively in fully-supervised settings and exhibits strong performance in the few-shot VideoQA setting we newly introduce.

**Limitations.** Promising directions not explored in this work include scaling the size of a bidirectional language model to several billion parameters, and additional training on large datasets of YouTube videos with accompanying speech transcripts and/or audio [110]. Also, our model cannot be applied out-of-the-box to complex multi-modal text generation tasks such as video captioning.

**Broader Impact.** We have showed the superior compute-efficiency of our bidirectional framework compared to autoregressive models for zero-shot VideoQA, and believe it is a step towards reducing the environmental impact of such research and its applications [84]. In addition, our models might reflect biases present in videos and captions from Shutterstock used to train our model, the text data used to train the language model or the images and captions used to train the visual backbone. It is important to keep this in mind when deploying, analysing and building upon these models.

**Acknowledgements.** This work was granted access to the HPC resources of IDRIS under the allocation 2022-AD011011670R2 made by GENCI. The work was funded by a Google gift, the French government under management of Agence Nationale de la Recherche as part of the "Investissements d'avenir" program, reference ANR-19-P3IA-0001 (PRAIRIE 3IA Institute), the Louis Vuitton ENS Chair on Artificial Intelligence, the European Regional Development Fund under project IMPACT (reg. no. CZ.02.1.01/0.0/0.0/15 003/0000468). We thank anonymous reviewers for giving interesting feedback. We thank Gaspard Beugnot, Clémence Bouvier and Pierre-Louis Guhur for proofreading.

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
