# OpenReview forum: "Zero-Shot Video Question Answering via Frozen Bidirectional Language Models"
_NeurIPS.cc/2022/Conference — NeurIPS 2022 Accept_

### Official Review · Reviewer_daCv · 2022-07-10

**Rating:** 6
**Confidence:** 4
**Soundness:** 3 good
**Presentation:** 3 good
**Contribution:** 3 good

**Summary:**

This paper mainly concerns the problem of zero-shot video question answering (VideoQA). It proposes FrozenBiLM, which performs video-text(-speech) pre-training over a frozen language model to enable it with visual understanding capability while preserving its language generalization ability. The resulted model could perform zero-shot VideoQA out of the box. The paper also demonstrates the capacity of FrozenBiLM in terms of few-shot/fully-tuned learning. New SotAs are set on a wide range of mainstream VideoQA benchmarks.

**Questions:**

Please address the following questions:

i) In line 217, it is evident that a simple averaging over multiple tokens does not preserve the semantic structure of the label/phrase (e.g., "man riding horse" vs. "horse riding man"). Hence, the resulted model could suffer in the wild when *rare* events/relationships happen (e.g., "horse riding man"). It would be great if the authors can shed some lights on possible solutions and perhaps other strategies they have attempted in the experiments to alleviate this issue. One of the related works is [r5].

ii) Considering that not all existing methods have speech transcripts as model input and the fact that transcripts are extremely informative esp. on instructional videos, the comparison seems unfair (e.g., in Tab. 6). Please comment on this and also denote methods with speech input in the result tables.

[r5] Salazar et al., Masked Language Model Scoring. ACL 2020.

**Limitations:**

Appears to be sufficient.

**Strengths And Weaknesses:**

Despite that adapting a frozen language model for VL tasks is not new (e.g., [88]), nor is adaptor-based transfer learning for Transformers (e.g., [26] and LoRA [r1]), the paper studies this simple yet important baseline on the challenging problem of VideoQA. The overall idea is convincing and well-executed. The experimental results are solid and adequately support the claims from the paper w.r.t. the importance of freezing the language model and adaptors on cross-domain transfer learning. The paper flows well and is easy to follow.

Note that some relevant concurrent work such as Flamingo [r2] is missing in the related work, considering its relevance and the similar model design. Other related work on zero-shot/few-shot VideoQA include [r3] [r4].

[r1] Hu et al., LoRA: Low-Rank Adaptation of Large Language Models. arXiv 2021.

[r2] Alayrac et al., Flamingo: a visual language model for few-shot learning. arXiv 2022.

[r3] Xu et al., Videoclip: Contrastive pre-training for zero-shot video-text understanding. EMNLP 2021.

[r4] Wang et al, Language Models with Image Descriptors are Strong Few-Shot Video-Language Learners. arXiv 2022.

---

> ### Author Response · Authors · 2022-08-02
> **Author response to Reviewer daCv (1/2)**
>
> We thank Reviewer daCv for providing constructive comments.
>
> > Despite that adapting a frozen language model for VL tasks is not new (e.g., [88]), nor is adaptor-based transfer learning for Transformers (e.g., [26] and LoRA [r1]), the paper studies this simple yet important baseline on the challenging problem of VideoQA.
>
> We wish to point out that bidirectional masked language models (BiLM) significantly differ from autoregressive language models [88] in terms of attention masking, training, and inference. As far as we know, our work is the first to explore the zero-shot capabilities of frozen BiLM for vision-language tasks.
>
> > Note that some relevant concurrent work such as Flamingo [r2] is missing in the related work, considering its relevance and the similar model design. Other related work on zero-shot/few-shot VideoQA include [r3] [r4].
>
> We thank reviewer daCv for providing these additional references that we will cite and discuss in the final version. We wish to kindly point out that [r2] and [r4] are concurrent works where [r4] was released on arXiv after the NeurIPS submission deadline. Both [r2] and [r4] leverage autoregressive language models for few-shot vision-language tasks while our work shows the benefit of using bidirectional masked language models (BiLM). We emphasize that BiLM significantly differ from autoregressive language models in terms of attention masking (bidirectional vs unidirectional), training (masked language modeling loss vs unidirectional language modeling loss with teacher forcing), and inference (predicting masked tokens vs left-to-right autoregressive generation). In particular, we argue that BiLM have sufficient natural language generation ability to perform VideoQA, and that their output can be better constrained to provide concise answers. Empirically, we have demonstrated that BiLM achieve superior performance and parameter efficiency compared to autoregressive language models, see Section 4.3.
> The work [r3] is related to ours but is focused on retrieval tasks and does not address video question answering. In particular, [r3] considers the multiple-choice video-to-text retrieval task of MSR-VTT as VideoQA. In this setting, the model is not provided with natural language questions. We will include this discussion to the related work section of the paper.

---

> > ### Comment · Reviewer_daCv · 2022-08-08
> > **Thanks for the responses**
> >
> > Thanks for the clarification and insights. The additional results and discussions on "Multiple mask tokens decoding" is thorough and convincing. The gain is indeed incremental; however, it's an interesting future direction for potential improvements. Please do include these additional results in the final version.

---

> ### Author Response · Authors · 2022-08-02
> **Author response to Reviewer daCv (2/2)**
>
> > i) In line 217, it is evident that a simple averaging over multiple tokens does not preserve the semantic structure of the label/phrase (e.g., "man riding horse" vs. "horse riding man"). Hence, the resulted model could suffer in the wild when rare events/relationships happen (e.g., "horse riding man"). It would be great if the authors can shed some lights on possible solutions and perhaps other strategies they have attempted in the experiments to alleviate this issue. One of the related works is [r5].
>
> To improve the modeling of multi-token answers of FrozenBiLM for open-ended VideoQA, we have taken inspiration from [A] and performed zero-shot VideoQA inference by using multiple mask tokens decoded in parallel. Then, for each video-question pair, we did one forward pass through the model per possible number of mask tokens (typically, 1 to 5) in order to score all possible answers in vocabulary A. The score of a given answer was then obtained by multiplying the probability of its individual tokens, possibly normalized by its number of tokens. We observed that such a decoding strategy did not significantly improve the accuracy of our model. This may be due to the fact that the current open-ended VideoQA datasets [29, 94, 96, 103] contain a great majority of short answers, e.g. 99% of the answers in the MSRVTT-QA test set are one-token long with our tokenizer. We report in Table T3 below detailed results with this inference strategy compared with the inference strategy used in the paper. Additionally, a possible solution to further improve the decoding in this alternative scheme is to increase the length of the masked spans at pretraining, as in [B]. We thank Reviewer daCv for suggesting a relevant reference [r5], which provides another potential solution to score multi-token answers in our framework. We will include this discussion in the paper.
>
> [A] Jiang et al., X-FACTR: Multilingual Factual Knowledge Retrieval from Pretrained Language Models, EMNLP 2020.
> [B] Joshi et al., SpanBERT: Improving Pre-training by Representing and Predicting Spans, TACL 2020.
>
> | Inference Strategy | LSMDC | iVQA | MSRVTT-QA | MSVD-QA | ActivityNet-QA | TGIF-QA |
> |:---:|:---:|:---:|:---:|:---:|:---:|:---:|
> | Average token embeddings | **51.5** | 26.8 | 16.7 | 33.8 | 25.9 | 41.9 |
> | Multiple mask tokens | 51.0 | **27.0** | **17.1** | **34.4** | **26.1** | **42.0** |
>
> Table T3 - Impact of the inference strategy on the zero-shot open-ended VideoQA performance.
>
> > ii) Considering that not all existing methods have speech transcripts as model input and the fact that transcripts are extremely informative esp. on instructional videos, the comparison seems unfair (e.g., in Tab. 6). Please comment on this and also denote methods with speech input in the result tables.
>
> As evaluated in Section 4.2, in the zero-shot setting, speech is particularly helpful on the How2QA and TVQA benchmarks, while it has a low impact on the other benchmarks. We will denote methods with speech input in Tables 5 and 6. In detail, no method uses speech as input in Table 5; in Table 6, the methods using speech as input are HCRN [42] on TVQA, HERO [51] on How2QA and TVQA, MERLOT [104] on TVQA, and RESERVE [105] on TVQA. In addition, in Tables T4 and T5 below we compare our method to state-of-the-art methods in the fair setup, when no speech input is used. These results show the effectiveness of our approach even when speech is not provided as input. We will include these results in Tables 5 and 6 in the paper.
>
> | Method | LSMDC | iVQA | MSRVTT-QA | MSVD-QA | ActivityNet-QA | TGIF-QA | How2QA | TVQA |
> |:---:|:---:|:---:|:---:|:---:|:---:|:---:|:---:|:---:|
> | SoTA | 31.0 [105] | 13.3 [97] | 5.8 [105] | 13.5 [97] | 12.3 [97] | 3.6 [68] | **53.1** [97] | 26.1 [68] |
> | FrozenBiLM (Ours) | **50.9** | **26.2** | **16.9** | **33.7** | **25.9** | **41.9** | 41.9 | **29.7** |
>
> Table T4 - Comparison to the state of the art for zero-shot VideoQA without speech as input.
>
> | Method | LSMDC | iVQA | MSRVTT-QA | MSVD-QA | ActivityNet-QA | TGIF-QA | How2QA | TVQA |
> |:---:|:---:|:---:|:---:|:---:|:---:|:---:|:---:|:---:|
> | SoTA | 53.7 [19] | 35.4 [97] | 46.8 [90] | 48.3 [90] | 41.4 [104] | **69.5** [104] | **85.3** [97] | 44.2 [52] |
> | FrozenBiLM (Ours) | **58.6** | **39.7** | **47.0** | **54.4** | **43.2** | 68.6 | 81.5 | **57.5** |
>
> Table T5 - Comparison to the state of the art for fully-supervised VideoQA without speech as input.

---

### Official Review · Reviewer_MVzv · 2022-07-11

**Rating:** 7
**Confidence:** 4
**Soundness:** 4 excellent
**Presentation:** 4 excellent
**Contribution:** 4 excellent

**Summary:**

The paper addresses zero-shot video question answering by leveraging pre-trained language models and visual encoders. The proposed model encodes videos with CLIP and projects them into language. The language is encoded with a bidirectional language model (e.g. BERT) and trained with a masked language modeling objective. The paper only re-trains the visual projection and adapter layers on a large video-captions dataset, and it shows that freezing the language model and the visual encoder leads to better results than re-training everything. Extensive experiments are conducted and results are reported on 8 videoQA datasets.

**Questions:**

- The drop in performance when removing the suffix is surprising. Does something similar happen when removing the prefix ([CLS])?
- How important is the prompt design for the zero-shot performance? Have you tried with different prompt templates?
- Have you tried using the model for image VQA?

**Limitations:**

Already discussed in the paper.

**Strengths And Weaknesses:**

- The paper is well-motivated, well-written, and easy to understand. The ideas proposed in the paper (joining pre-trained models together and use them for zero-shot videoQA without retraining) are simple and elegant. Multiple experimental results and ablations on 8 datasets support the hypothesis of the paper.

- The results of the paper are significant to the community:
  1. Using frozen language models (with no visual information training at all) can lead to better performance than fine-tuning such models on visual-language datasets probably due to catastrophic forgetting.
  2. Smaller language models like BERT and its variants can outperform larger language models based on GPT.
  3. Models pre-trained for different tasks can be made to work together by only fine-tunning some adapter layers.

---

> ### Author Response · Authors · 2022-08-02
> **Author response to Reviewer MVzv**
>
> We thank Reviewer MVzv for providing a thoughtful review.
>
> > The drop in performance when removing the suffix is surprising. Does something similar happen when removing the prefix ([CLS])?
>
> We hypothesize that the suffix helps the model to provide a concise answer, as the suffix is placed next to the right of the mask token that is used to predict the answer. We do not observe an important drop in performance when removing the [CLS] token during zero-shot VideoQA inference, e.g., the accuracy on MSVD-QA slightly drops from 33.8% to 33.2%.
>
> > How important is the prompt design for the zero-shot performance? Have you tried with different prompt templates?
>
> To further investigate the prompt design, we have explored replacing the words “Question”, “Answer” and “Subtitles” by “Q”, “A” and “S”, respectively, in the templates described in Section 3.3. This change did not impact the zero-shot VideoQA accuracy, however, completely removing “Question”, “Answer”, “Subtitles” and “is it” in the templates resulted in a significant drop of performance. We report detailed results in Tables T1 and T2 below. We conclude that it is important to have tokens that link the different textual inputs. We will add and discuss these results in the final version of the paper.
>
> | Template | iVQA | MSRVTT-QA | MSVD-QA | ActivityNet-QA | TGIF-QA |
> |:---:|:---:|:---:|:---:|:---:|:---:|
> | [CLS] Question: <Question>? Answer: [MASK]. Subtitles: <Subtitles> [SEP] | 26.8 |  **16.7** | **33.8** | **25.9** | **41.9** |
> | [CLS] Q: <Question>? A: [MASK]. S: <Subtitles> [SEP] | **27.4** | 16.2 | 32.5 | 25.5 | **41.9** |
> | [CLS] <Question>? [MASK]. <Subtitles> [SEP] | 23.1 | 13.6 | 28.0 | 21.6 | 25.2 |
>
> Table T1 - Impact of the prompt on zero-shot open-ended VideoQA performance.
>
> | Template | How2QA | TVQA |
> |:---:|:---:|:---:|
> | [CLS] Question: <Question>? Is it ’’<Answer Candidate>”? [MASK]. Subtitles: <Subtitles> [SEP]” | **58.4** | **59.7** |
> | [CLS] Q: <Question>? Is it ’’<Answer Candidate>”? [MASK]. S: <Subtitles> [SEP]” | 57.7 | 58.2 |
> | [CLS] <Question>? <Answer Candidate>? [MASK]. <Subtitles> [SEP] | 47.6 | 55.0 |
>
> Table T2 - Impact of the prompt on zero-shot multiple-choice VideoQA performance.
>
> > Have you tried using the model for image VQA?
>
> We have evaluated our pretrained model on the VQAv2 validation set in the zero-shot setting, i.e., without any supervision of visual questions and answers. Frozen [88] achieves 29.5% accuracy in this setting using an autoregressive language model. In comparison, our FrozenBiLM model is 7 times smaller than Frozen and achieves 45.0% accuracy. We conclude that our model can perform competitively on the image-VQA tasks despite being tailored for videos. We will add these results to the final version of the paper.

---

> > ### Comment · Reviewer_MVzv · 2022-08-09
> > **Response**
> >
> > Thank you for the detailed response and the extra experiments on the template design and image VQA. After reading all the reviews and discussions, I am happy to support this paper for acceptance.

---

### Official Review · Reviewer_bWDb · 2022-07-11

**Rating:** 6
**Confidence:** 4
**Soundness:** 4 excellent
**Presentation:** 4 excellent
**Contribution:** 3 good

**Summary:**

This paper proposed to leverage the power of a frozen bidirectional LM to tackle the zero-shot video question-answering task. Adapters and projection layers are added to the model as the only trainable weights, so it's a very lightweight optimization. Also, the frozen backbones enable us to fine-tune the model without forgetting the knowledge of the pretrained backbones, so it can achieve much better performance compared to fine-tuning the whole model. The proposed model achieves state-of-the-art performance on many zero-shot VideoQA datasets. The experiments in the fully-supervised setting also show that it's possible to apply the same method with much training data and achieve state-of-the-art results.

**Questions:**

- Is it possible to adapt a pretrained speech module to this framework? As audio is crucial information in videos. Merely adding ASR transcripts to the model can not fully preserve the audio information. If we can incorporate pretrained models like wav2vec, this direction would be much more interesting.

**Limitations:**

The authors adequately addressed the limitations and potential negative societal impact in the Conclusion section.

**Strengths And Weaknesses:**

Strengths:
- The paper is well-written. The flow is easy to follow and the method is clear and simple.
- The authors carefully design the prompting templates for using the frozen model out-of-the-box.
- The ablation study is comprehensive, we can clearly see that a frozen LM and adapters and the speech modality are necessary to to achieve good performance.
- The result shows the effectiveness of this proposed method: it can outperform a fine-tuning baseline or an autoregressive baseline by a large margin.
- The same method can also be applied to the fully-supervised setting and it can achieve state-of-the-art results when compared with other supervised baselines.

Weaknesses:
- The idea of utilizing a frozen LM module is not new (e.g. https://arxiv.org/abs/2106.13884 which is a paper using autoregressive LMs to perform image-text few-shot learning). Although this method mainly focuses on VideoQA and BiLM, its novelty is still limited. However, this paper has a good engineering value, as no one has tried to use a frozen BiLM on VideoQA tasks. This paper is the first to do that can achieve surprisingly good results.

---

> ### Author Response · Authors · 2022-08-02
> **Author response to Reviewer bWDb**
>
> We thank Reviewer bWDb for providing feedback.
>
> > The idea of utilizing a frozen LM module is not new (e.g. https://arxiv.org/abs/2106.13884 which is a paper using autoregressive LMs to perform image-text few-shot learning). Although this method mainly focuses on VideoQA and BiLM, its novelty is still limited. However, this paper has a good engineering value, as no one has tried to use a frozen BiLM on VideoQA tasks. This paper is the first to do that can achieve surprisingly good results.
>
> Using frozen autoregressive language models for few-shot/zero-shot vision-language tasks has indeed been previously explored [17, 65, 88, 99], as acknowledged in Section 2. However, we wish to emphasize that bidirectional masked language models (BiLM) significantly differ from autoregressive language models in terms of training and inference (predicting masked tokens vs left-to-right autoregressive generation). To accommodate for these differences, we have designed a specific inference strategy tailored for zero-shot VideoQA using BiLM, see Section 3.3. As far as we know, our work is the first to explore the zero-shot capabilities of frozen BiLM for vision-language tasks. We hope that our work raises more interest in exploiting BiLM for few-shot/zero-shot learning of vision and language tasks, in analogy to the literature dedicated to the few-shot capabilities of BiLM in natural language processing [62, 73, 74, 84]. In addition, we also demonstrate the benefits of freezing a BiLM in supervised settings, while current fully-supervised state-of-the-art approaches for VideoQA typically train a BiLM end-to-end [19, 45, 51, 90, 97, 104, 105]. Finally, we also evaluate the video-conditioned fill-in-the-blank task which is challenging for autoregressive language models, and show we can achieve state-of-the-art results on LSMDC.
>
> > Is it possible to adapt a pretrained speech module to this framework? As audio is crucial information in videos. Merely adding ASR transcripts to the model can not fully preserve the audio information. If we can incorporate pretrained models like wav2vec, this direction would be much more interesting.
>
> Audio is indeed a valuable source of information in videos that goes beyond speech transcripts. For example, hearing a camel grunting in the fourth example of Figure 3 could help answering the question. Extending our FrozenBiLM model with audio input is certainly possible as our architecture is agnostic to the added modality (here vision) to the language model. For example, one could linearly project features extracted from a pretrained audio encoder (e.g. wav2vec) to the token embedding space before feeding them to the frozen language model. Note that we would also need a pretraining dataset with videos that contain audio like HowTo100M [64] or YT-Temporal-1B [104, 105] instead of the audio-less WebVid10M dataset used in our work. We leave this interesting direction to future work.

---

> > ### Comment · Reviewer_bWDb · 2022-08-08
> > **Thanks for the responses**
> >
> > Thanks for responding to my questions. I agree that BiLM significantly differs from autoregressive language models in terms of training and inference. And I am looking forward to seeing the future work of incorporating the speech part. I am inclined to let this paper be accepted.

---

### Author Response · Authors · 2022-08-02
**Author response - Overview**

We thank the reviewers for their helpful comments. We appreciate encouraging remarks that our idea of tackling zero-shot video question answering with frozen bidirectional language models is elegant (MVzv) and convincing (daCv). Reviewers also found that our paper is well written and flows well (bWDb, MVzv, daCv). They acknowledged that our experiments adequately support the claims from the paper (MVzv, daCv) and that our results are significant to the community (MVzv). These results include a comprehensive ablation study (bWDb) and state-of-the-art results on a wide range of zero-shot/fully-supervised video question answering benchmarks (bWDb, daCv).

We provide detailed answers to the comments and questions from each reviewer in the different author responses and will modify our paper accordingly.

---

### Meta-Review · Area_Chair_XToR · 2022-08-29

**Recommendation:** Accept
**Confidence:** Less certain

**Metareview:**

The submission introduces a zero-shot VQA model that combines frozen video and bidirectional language models by training additional projection and adaptor layers. The method significantly outperforms related previous work that uses only uni-directional language models. While the approach is somewhat incremental technically, reviewers found the results to be significant and thought that the main claims of the paper are well supported by thorough ablations. There are no significant concerns raised by the reviews, and overall this is solid work, so I recommend acceptance.

**Award:**

No

---

### Decision · Program_Chairs · 2022-09-14

Accept